# ⊞DrawEduMath: Evaluating Vision Language Models with Expert-Annotated Students' Hand-Drawn Math Images

**Sami Baral**[□*]   **Li Lucy**[△*]
**Ryan Knight**[+]   **Alice Ng**[=]   **Luca Soldaini**[÷]
**Neil T. Heffernan**[□]   **Kyle Lo**[÷]

[□]Worcester Polytechnic Institute   [△]University of California Berkeley
[+]Insource Services   [=]Teaching Lab   [÷]Allen Institute for AI

## Abstract

In real-world settings, vision language models (VLMs) should robustly handle naturalistic, noisy visual content as well as domain-specific language and concepts. For example, K-12 educators using digital learning platforms may need to examine and provide feedback across many images of students' math work. To assess the potential of VLMs to support educators in settings like this one, we introduce ⊞DrawEduMath, an English-language dataset of 2,030 images of students' hand-written responses to K-12 math problems. Teachers provided detailed annotations, including free-form descriptions of each image and 11,661 question-answer (QA) pairs. These annotations capture a wealth of pedagogical insights, ranging from students' problem-solving strategies to the composition of their drawings, diagrams, and writing. We evaluate VLMs on teachers' QA pairs, as well as 44,362 synthetic QA pairs derived from teachers' descriptions using language models (LMs). We show that even state-of-the-art VLMs leave much room for improvement on ⊞DrawEduMath questions. We also find that synthetic QAs, though imperfect, can yield similar model rankings as teacher-written QAs. We release ⊞DrawEduMath to support the evaluation of VLMs' abilities to reason mathematically over images gathered with educational contexts in mind.

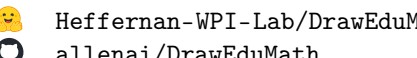

🤗 Heffernan-WPI-Lab/DrawEduMath
○ allenai/DrawEduMath

## 1   Introduction

As AI models demonstrate growing proficiency in mathematical reasoning, there is a corresponding rise in AI-powered tools designed to enhance math education [21, 12, 14, 34]. For example, AI systems have the potential to provide immediate feedback on students' work [5], or shed insight on common misconceptions [16]. These trends prompt critical questions about the ability of current models to handle real-world math problems, such as those encountered in classrooms and tutoring sessions, as opposed to curated problems found in popular benchmarks like GSM8k [8] and MATH [19]. We present ⊞**DrawEduMath**, a collection of 2,030 images of K-12 math problems paired with images of *handwritten, hand-drawn responses* to these problems by real student users of an online learning platform. This collection encompasses a diverse array of mathematical concepts, educational standards, and problem types. We supplement all images with the following:

---

[*]Both authors contributed equally to this research.
    Contact: {sbaral,nth}@wpi.edu, lucy3_li@berkeley.edu, rknight@insourceservices.com, alice.ng@teachinglab.org, {lucas,kylel}@allenai.org

38th Conference on Neural Information Processing Systems (NeurIPS 2024).

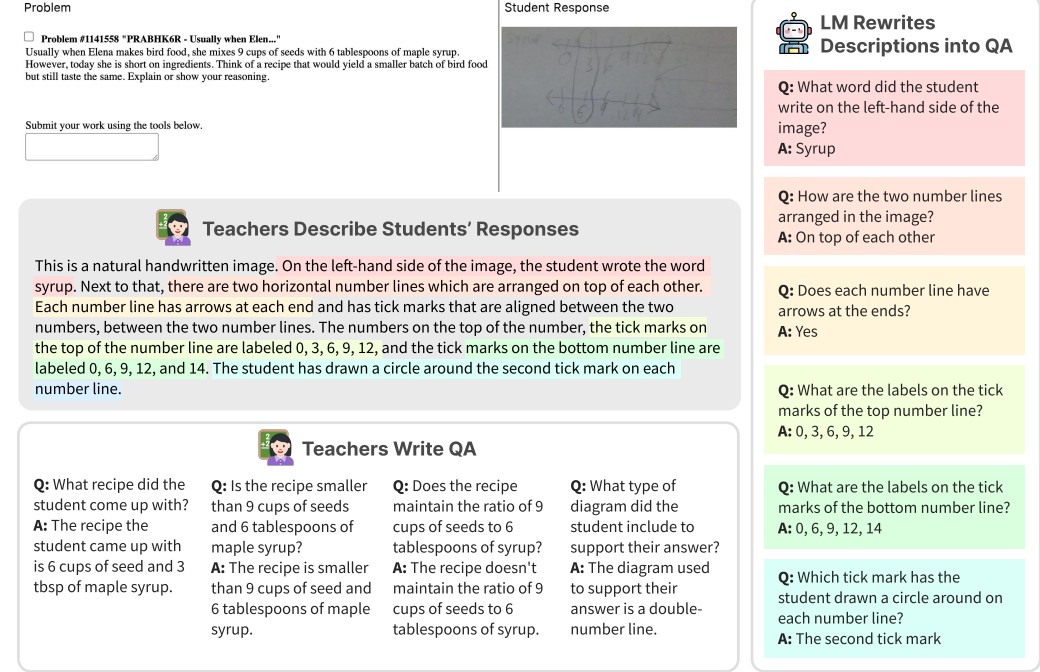

Figure 1: Each image in our dataset is a concatenation of a math problem on the left with a student response on the right. Teachers describe the student's response to the problem, and then a model, such as GPT-4o shown here, writes QA pairs extracted from facets of the description. More example images, along with teacher-written QA, are shown in Figure 3.

1. **Detailed descriptions** provided by teachers, capturing all elements of the student's handwritten responses, including the students' approach, possible misconceptions, and mistakes made during problem-solving.
2. **Question-answer (QA) pairs**, some of which are written by teachers and some generated through an LM-based pipeline. The latter involves identifying key facets in teachers' descriptions and restructuring them into questions and answers.
3. **Metadata** for each image, encompassing the type of problem, corresponding educational standards or grade level, topical categories, and other relevant information.

In this work, we detail our benchmark creation process (§3), which aims to balance educators' expertise and the scalability of LM-based data generation and judgement (§4). We then use ⊞DrawEduMath to evaluate the capabilities of current VLMs to interpret the content of students' handwritten responses (§6). We find that though models can identify superficial aspects of images such as paper type and drawing medium, they struggle on questions related to the correctness of students' responses. In addition, closed models such as Claude and GPT-4o tend far outperform open-source Llama 3.2-11B. Overall, we hope that this work will facilitate further research on VLMs' abilities to support students' math learning in diverse, real-world educational settings.

## 2 Related Work

**AI for Math Education.** The advent of large language models (LLMs) has transformed online learning platforms [1, 11, 44, 18] by introducing automated tools for error identification [16, 43, 37], feedback provision [30], student response scoring [4], and curriculum adaptation [28], primarily for typed answers. However, most math instruction in traditional classrooms still relies on handwritten problem-solving, posing challenges due to the unstructured nature of handwritten content and a lack of annotated datasets [3]. Existing math datasets, such as GSM8k [8] or MATH [19], focus on K-12 content but often lack input from educators, leaving a gap in aligning AI research with the classroom realities. While the recent advancements in multimodel LLM capabilities allow for the interpretation of complex images [46], their effectiveness in understanding student handwritten math

| Math Domain | Images | Example Words or Phrases in Teachers' Annotations of Images |
|---|---|---|
| Ratios & Proportions | 29.9% | *proportional relationship, cups, proportional reasoning, 4x, equivalent ratios, corresponding values, scoops, double number, multiplicative relationship, proportional line* |
| Geometry | 24.4% | *xyz, x'y'z', isosceles triangle, perpendicular bisector, rigid transformation, equilateral triangle, original triangle, two quadrilaterals, equilateral triangles, original image* |
| Expressions & Equations | 14.7% | *negative infinity, connected rectangles, x+1, 5x, x., number line, arrow pointing, horizontal rectangle* |
| The Number System | 9.5% | *vertical number, shaded sections, five sections, negative integers, negative numbers, algorithm subtraction, incorrect representation, positive numbers, rectangular model, division algorithm* |
| Number & Operations, Fractions | 6.6% | *fraction strips, whole numbers, fractional parts, rectangular fraction, equivalent fractions, mark, identical rectangles, horizontal rectangle, equivalent fraction, tick* |

Table 2: The top five most frequent math domains, as defined by CCSS, that appear ⊞DrawEduMath. Example words or phrases were obtained by applying the `phrasemachine` text analysis tool [17] on teachers' descriptions and answers. The examples shown have the highest TF-IDF scores within each domain and occur across at least two problems' images. Percentages show the relative frequency of each domain across all annotated images.

remains uncertain. This paper aims to address this gap by contributing a benchmark created by real students and teachers.

**Vision-language Evaluation and Benchmarks.**    The growth of pretrained VLMs accompanies the growth of vision-language benchmarks, e.g. MMMU [45], DocVQA [32], and VQA [15]. Within the domain of math, notable examples include MathVista [25], GeoQA [7], Geometry3k [26], and MathVerse [46]. Many of these prior visual math benchmarks, however, focus on images where mathematical information is shown in a standardized or typed manner. In contrast, the images in our dataset consist mostly of handwriting and drawings across different paper, lighting, and digitization types. In addition, our focus on problem solving strategies and pedagogy allows our annotations to go beyond optical character recognition emphasized in previous handwritten datasets [9, 29, 24, 48, 35, 31, 13].

# 3    The ⊞DrawEduMath Dataset

Our dataset begins by sampling images of K-12 students' responses to math problems, followed by two rounds of annotation by teachers. During annotation, we ask teachers to both describe students' responses and write a few QA pairs for each image. Overall, teachers' annotations mention a variety of K-12 mathematical concepts and representations (Table 2). In total, this process yields 2,030 described images and 11,661 teacher-written QA pairs (Table 3, Table 6).

## 3.1    Sampling Students' Math Images

Our dataset consists of 2,030 images of U.S.-based students' handwritten math responses to 188 math problems spanning Grade 2 through high school (Table 1). These images were initially collected on the ASSISTments [18] online learning platform, where students receive feedback from teachers on assigned work. The problems that accompany each student response are drawn from three overlapping[1] open educational resources (OER): Eureka Math, Open Up

| 📝 Students' Math Images | |
|---|---|
| # of annotated images | 2,030 |
| # of math problems | 188 |
| Avg # of images per problem | 12.64 |
| % of problems in Grades 2-5 | 34.6 |
| % of problems in Grades 6-8 | 81.4 |
| % of problems in High School | 10.1 |
| # of math standards covered | 86 |
| # of math domains covered | 12 |

Table 1: Key data statistics pertaining to students' math images included in ⊞DrawEduMath.

Resources, and Illustrative Math. Metadata linked to these problems include Common Core State Standards (CCSS) labels, which indicate specific K-12 math skills or concepts targeted in problems [39]. Initially, the data provided by the learning platform comprised approximately 60,000 images across 188 problems, with an average of 300 images per problem. From this, we randomly sampled 15 images per problem. To ensure student privacy, undergraduate research assistants cropped the images to include only the math content and removed any personally identifiable information, such as students' hands, by covering them with dark rectangles. Our use of these images was deemed exempt from review by our institution's institutional review board; see more discussion in §9.

---

[1]OER materials may reuse or adapt problems from each other; hence, some problems in our dataset appear across more than one content source.

## 3.2 Collecting Teachers' Annotations

We hired three NYC-based math teachers from Teaching Lab, a nonprofit professional learning organization, to describe each image. We paid teachers over $200 USD per hour. Each teacher had at least 6 years of experience in math education, with two teachers specializing in middle school and one teacher in grades 5-12. Teachers annotated images on a custom website, and were asked to describe an image as thoroughly as possible so that another teacher could recreate it without viewing it. The annotation website presented an image concatenating the original problem with a student's response, followed by a text box for typed notes and a speech recording module. Teachers also noted whether an image is too blurry for annotation and flagged any PII, adding an extra security layer to our initial PII removal process §3.1.

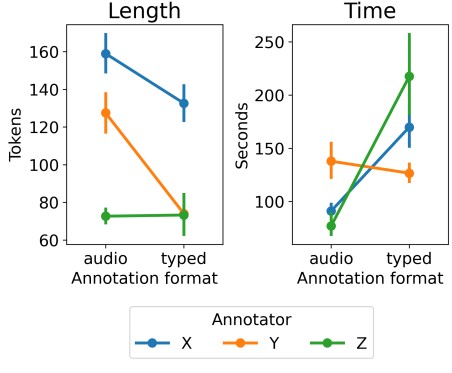

Figure 2: For some annotators, their recorded descriptions of images are longer or require less time than typed ones. Annotation length is calculated based on white-spaced-separated tokens.

Some annotations were obtained by transcribing recordings of teachers' spoken descriptions, while others were typed into an text box. We offered the option of both annotation modalities because spoken descriptions are sometimes faster to obtain and result in longer annotations [38, 10], but typing gives teachers the flexibility to annotate in noisy environments and reduces the risk of transcription errors; see comparison in Figure 2. We obtained similar amounts of typed and recorded image descriptions (Table 3). Full annotation instructions, a screenshot of our setup, and additional details on our data collection process can be found in Appendix A.1.

Over the course of two months, teachers annotated 2,376 images of students' responses. After removing images that were deemed too blurry or failed a secondary PII check, our final dataset consists of 2,030 images paired with math teachers' descriptions.

## 3.3 Revising and Augmenting Annotations

During a second data collection phase, teachers augmented and revised existing annotations. This second phase of annotation required twice as much time per example than the first one (Table 3). So, to complete this phase, we recruited five additional teachers from the same professional learning organization as we did in §3.2. Each of these additional teachers had at least 9 years of experience in math education spanning the UK and several U.S. states, including two from the NYC area. Grade level expertise among these five teachers include one in 9-12, one in 5-12, two in K-8, and one in K-12.

| 🧑‍🏫 Teachers' Annotations | |
|---|---|
| *First round* | |
| Avg minutes spent per image | 2.0 |
| Total words in descriptions | 228k |
| Avg description length | 111.1 |
| % of descriptions typed | 46.7 |
| % of descriptions transcribed | 53.3 |
| *Second round* | |
| Avg minutes spent per image | 4.3 |
| Total words in descriptions | 222k |
| Avg description length | 109.5 |
| % of descriptions left unchanged | 94.2 |
| Median edit distance of changed descriptions | 48.5 |
| # of teacher-written QA pairs | 11,661 |
| Avg # of teacher-written QA per image | 5.74 |
| Avg length of teacher-written questions | 12.7 |
| Avg length of teacher-written answers | 16.2 |

Table 3: Key data statistics pertaining to the collection of teachers' language for 📊DrawEduMath. Word counts and text lengths are determined using white-space delineated tokens.

**Revising Teachers' Initial Descriptions.** During reannotation, teachers were allowed to revise the image's description, to correct possible transcription errors or other clarity issues that arose during initial annotations. The vast majority (>90%) of image descriptions were not edited, and when edits were made, the Levenshtein distance between old and new descriptions was typically small (Table 3). Through qualitative inspection of edits, most were typo corrections, e.g. *rose → rows* or *three four → three fourths*.

**Adding Teacher-written QA.** The main part of our second annotation round focuses on augmenting descriptions with questions teachers may ask about students' responses. We asked teachers to come up with questions that they would naturally ask when examining student responses and were provided with example topics, such as whether the student demonstrated a mathematical concept or made

**What errors does the student make in their response?**

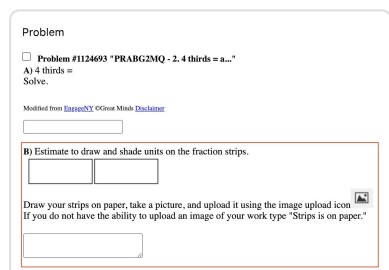
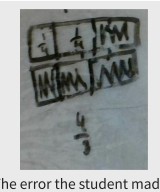
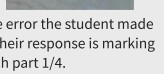
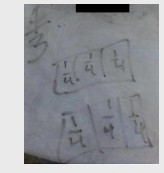
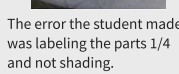
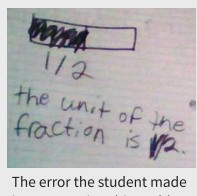

The error the student made in their response is marking each part 1/4.

The error the student made was labeling the parts 1/4 and not shading.

The error the student made is representing this problem with 1/2.

Figure 3: Examples of teacher's answers to a question asking about possible errors in students' responses to math problems. All three examples of students' hand-drawn responses are for the same math problem asking students to draw and shade units on fraction strips to show 4 thirds, shown on the left.

a common error for a problem type. This top-down data collection approach in this second round complements the bottom-up, description-based approach emphasized in the first round §3.2, and may cater more towards potential uses of VLM-based systems for educators.

First, teachers propose questions based on math problems in our dataset. Given a problem, teachers write up to five questions they may have about any student's response to that problem (Figure 1). Then, we present teachers with images of students' responses annotated in §3.2, and ask them to write answers to each problem-specific question based on what they observe in each student's response. Two additional questions, *What errors does the student make in their response?* and *What strategy does the student use to solve the problem?* were answered for all problems and student responses (Figure 3), and teachers also had the option to add up to two additional image-specific question-answer pairs. Across all 2,030 images, teachers augmented our ⊞DrawEduMath with 11,661 QA pairs.

## 4  Scaling Data with Synthetic QAs

Writing numerous QA pairs for visual benchmark creation is more time-intensive than describing images in a free-form manner (Table 3). Inspired by [6], who introduce a scalable workflow for generating VQA benchmarks from image captions, we use LMs to transform teachers' descriptions into synthetic QAs.

**Transforming Descriptions to QA Pairs.** We prompt Claude-3.5 Sonnet and GPT-4o to first decompose captions into "facets", or atomicized snippets of information, and rewrite these facets into question-answer (QA) pairs [6] (Figure 1). The prompts were iteratively refined with input from an expert teacher to enhance the quality of the generation responses. Specifically, the models were instructed to generate self-contained facets and corresponding QA pairs, avoiding open-ended questions or those with multiple correct answers. The full prompt we used for this data transformation step can be found in Appendix B.

| 🤖 Descriptions → 🧑 Synthetic QA pairs | |
|---|---|
| # of Claude-generated QA pairs | 21,089 |
| Avg # of Claude's QA per image | 10.3 |
| Avg length of Claude's questions | 10.6 |
| Avg length of Claude's answers | 2.2 |
| # of GPT-4o-generated QA pairs | 23,273 |
| Avg # of GPT-4o's QA per image | 11.5 |
| Avg length of GPT-4o's questions | 10.4 |
| Avg length of GPT-4o's answers | 3.0 |

Table 4: Key data statistics pertaining to synthetic QA pairs in ⊞DrawEduMath. Word counts for determining lengths are based on white-space delineated tokens.

We obtain a total of 44,362 synthetically created QA pairs (Table 4). On average, LM-generated QA had much shorter answers than those written by teachers, due to instructions preferring conciseness included in our description-to-QA prompt. Shorter answers are more suitable for reference-based evaluation with lightweight metrics such as string or ngram matching, but longer answers by teachers contain more rich and detailed information.

**Quality Assessment of Synthetic QA.** Two annotators examined a sampled set of QA pairs outputted from our description-to-QA pipeline to assess their quality. These annotators have complementary backgrounds, both of which are valuable for examining the application of VLMs for education: one has worked as a K-12 math teacher (Evaluator $\mathcal{A}$), and another has worked on technology applications for educators (Evaluator $\mathcal{B}$). For each image and QA pair, we ask: 1) *Can*

*this question be answered by the provided image?* and 2) *Is the provided answer correct?* 100 QA pairs were randomly sampled, evenly split between GPT-4o and Claude 3.5, with annotators each reviewing 50 pairs. Instructions for synthetic QA assessment can be found in Appendix D.1.

Despite some variability in annotators' judgments, the majority of QA pairs are answerable and correct (Table 5).

From qualitative inspection, unanswerable questions tend to be those where the referent of mentions is ambiguous without additional context. For example, a question may ask, *Where does the second arrow point?*, but it may be unclear which of the overlapping arrows in the image is the "second" one. So, "unanswerability" relates to the extent to which one infers ambiguous referents through pragmatic convention; for example, the *first piece* in a row of rectangles may be the one furthest left, and the *first*

| | Can this Q be answered? | | Is the provided A correct? | |
|---|---|---|---|---|
| | $\mathcal{A}$ | $\mathcal{B}$ | | $\mathcal{A}$ | $\mathcal{B}$ |
| Yes | 50 | 41 | Yes | 47 | 43 |
| No | 0 | 9 | No | 3 | 7 |

Table 5: Quality assessment of questions (Q) and answers (A) extracted by Claude & GPT-4o from teachers' descriptions of students' responses.

*triangle* in a geometric transformation may be the preimage. As for incorrect answers, Evaluator $\mathcal{B}$ marked some answers as incorrect due to the question being unanswerable. A few incorrect answers emerged from what appeared to be genuine annotation mistakes. For example, in one case, the annotator excluded the label on one tick mark in their annotation, and so the extracted QA's answer missed one value. Overall, we hope our inclusion of teachers' original descriptions in ⊞DrawEduMath can facilitate future improvements to the scaling of VQA benchmark creation.

## 5 Building a Taxonomy of Question Types

To document what types of questions show up in ⊞DrawEduMath and better understand which questions may be more difficult for models than others, we group questions into several categories. We defined question categories in an iterative manner mixing qualitative and quantitative approaches, akin to [36], who reframe content analysis into pattern detection, refinement, and confirmation steps. During pattern detection, we qualitatively code a combined pool of generated and teacher-written questions. To efficiently observe a range of common yet distinctive question patterns during this coding step, we sampled ten questions from clusters of questions' sentence embeddings [42].[2] We obtained these clusters using $k$-means with $k$=30, and embed questions after masking out their nouns,[3] so that we can examine problem-agnostic question patterns shared across different math domains. For example, questions that start with *How many...*, *Into how many...*, and *What is the total...* would occur in the same embedding cluster.

Next, for category refinement and confirmation, we recoded our observations into possible question types for GPT-4o to categorize. We iterated over question types and categorization prompts by running GPT-4o on smaller samples of 500 to 2000 questions. Proposing more fine-grained or more numerous question categories led to less cleanly delineated outputs, and so we aimed for category definitions that led to reasonable groupings. Our final prompt can be found in Appendix B.

Our resulting taxonomy of questions separates them into seven categories: 1) higher-level understanding of math content, 2) low-level content composition & positioning, 3) writing & labels, 4) problem solving steps, strategy, & solution, 5) counting content, 6) image creation & medium, and 7) correctness & errors (Table 6). In particular, the first two categories are designed to separate out questions that involve some mathematical reasoning from those that do not. For example, *What is the slope of the line* requires knowing what a slope is and how it's depicted in a graph, while questions that differentiate left from right pertain to more basic spatial understanding.

As shown in Table 6, (1) we find little difference in QA generation behavior between our two choices of LM, and (2) teachers' questions focus more on students' problem-solving steps and response correctness, while synthetic questions have a different emphasis.[4] An eighth category, "Other", which

---

[2]Specifically, the `all-mpnet-base-v2` embedding model.

[3]Nouns were detected using a spaCy part-of-speech tagger.

[4]The percentages for teacher QA shown in Table 6 do not include the two questions answered across all images.

| Question Type | Claude | GPT-4o | Teacher | Examples |
|---|---|---|---|---|
| Higher-level understanding of math | 26.7% | 25.7% | 18.8% | *What type of mathematical representation has the student drawn on the paper? What is the slope of the line passing through (0,-5) and (4,-4)? Is the student's image a third or a half of the original ratio to get 1 batch of light yellow paint?* |
| Low-level composition and positioning | 21.9% | 20.0% | 11.4% | *In the third row, where does the student place the number 3? Does the tens place in 15,420 line up beneath the tens place in 1542? Are the two pieces in the student's tape diagram equal or unequal in size?* |
| Writing and labels | 14.6% | 16.1% | 17.3% | *What number is written in front of Pam's rectangle, after the label 'Pam'? What range of numbers is labeled on each number line? What did the student label the top of the rectangle?* |
| Problem solving steps, strategy, and solution | 9.2% | 10.5% | 23.2% | *How does placing 26 directly above 25 help the student? Does the student start solving the problem with exact calculations or estimations? What method is the student using to prove that 3/50 equals 0.06?* |
| Counting content | 10.5% | 9.1% | 5.7% | *What is the total number of shaded-in pieces? How many tick marks are in between 2 and 3? How many rows and columns does the array have?* |
| Image creation and medium | 15.0% | 16.0% | 0.0% | *Is the student work drawn on graph paper or blank paper? On what surface is the image drawn? Are both triangles in the image pre-printed or is one drawn by the student?* |
| Correctness and errors | 1.7% | 1.5% | 23.0% | *Does the student get the correct or incorrect answer when adding 30 and 15 together? Did the student keep track of where all the vertices are supposed to be after rotation? Did the student correctly apply the scale factor of 1/2?* |

Table 6: The most common question types in our visual QA benchmark, along with examples of questions categorized within each type. The percentages shown are the proportion of questions across all images within each QA-writing (Claude-generated, GPT-4o-generated, or teacher-written) workflow.

we asked GPT-4o to use if a question fits into none of the provided categories, only makes up 0.4%, 1.1%, 0.6% of Claude, GPT-4o, and teacher-written questions, respectively.

# 6  Evaluating Vision Language Models with ▦DrawEduMath

| | GPT-4o QA | | | | Claude QA | | | | Teacher QA | | | |
|---|---|---|---|---|---|---|---|---|---|---|---|---|
| Model | BERT | ROUGE-L | LLM | Human (n=31) | BERT | ROUGE-L | LLM | Human (n=31) | BERT | ROUGE-L | LLM | Human (n=63) |
| GPT-4o | 0.835 | **0.544** | **0.700** | 0.742 | 0.843 | 0.599 | **0.743** | 0.677 | 0.752 | 0.199 | 0.628 | 0.524 |
| Claude 3.5 Sonnet | **0.856** | 0.537 | 0.697 | **0.871** | **0.883** | **0.608** | 0.732 | **0.742** | 0.754 | 0.202 | **0.657** | **0.587** |
| Gemini 1.5 Pro | 0.815 | 0.461 | 0.627 | 0.774 | 0.826 | 0.514 | 0.665 | 0.581 | 0.711 | 0.118 | 0.490 | 0.365 |
| Llama 3.2-11B V | 0.731 | 0.174 | 0.368 | 0.387 | 0.729 | 0.176 | 0.408 | 0.323 | **0.785** | **0.253** | 0.296 | 0.127 |

Table 7: Overall evaluation results for models across different VQA datasets generated by GPT4o, Claude, and human teachers. The table presents evaluations using automated metrics (BERTSCORE, ROUGEL), as well as assessments from LLMs and human evaluators. **Bold** is the max score across each metric.

**Experimental Setup.**    To assess the capability of recent visual language models (VLMs) in interpreting students' handwritten math work, we run several VLMs on ▦DrawEduMath. We experiment with four VLMs: three commercial models—GPT-4o, Claude 3.5 Sonnet [2], and Gemini 1.5 Pro [41]—alongside open-source Llama 3.2-11B Vision [33]. To select a prompt for running our experiments, we iterated over three possible prompts for each model on samples of data and selected the best-performing prompt across them. Our final prompt asks a model to succinctly answer a given question based on the student's response in a provided image (Appendix C).

**Automatic Evaluation.**    To compare VLMs' answers against gold ones, we employ three automatic metrics: (i) ngram matching via ROUGE-L [23], (ii) answer embedding similarity via BERTSCORE[5] [47], and (iii) LLM-based similarity judgements using Mixtral 8x22B [20]. Our prompt for the latter can be found in Appendix C, and asks models to rate the level of similarity between two answers given a question on a scale of 1 (*Quite different answers)* to 4 (*Basically the same*). When reporting results, we binarize these outputs so that 1-2 is counted as incorrect, and 3-4 are counted as correct.

**Human Evaluation.**    To validate our use of reference-based automatic metrics, 5 authors annotated a random sample of 500 QA responses, where 50% are teacher-written QA, 25% are Claude-generated

---

[5]With `distilbert-base-uncased` embedding model.

| Question Type | GPT-4o | | Claude 3.5 Sonnet | | Gemini 1.5 Pro | | Llama 3.2-11B V | |
|---|---|---|---|---|---|---|---|---|
| | 🤖 | 👩‍🏫 | 🤖 | 👩‍🏫 | 🤖 | 👩‍🏫 | 🤖 | 👩‍🏫 |
| Correctness & errors | 0.525 | 0.559 | 0.491 | 0.610 | 0.601 | 0.440 | 0.402 | 0.276 |
| Counting content | 0.642 | 0.671 | 0.516 | 0.667 | 0.602 | **0.578** | 0.247 | 0.265 |
| Higher-level understanding | 0.696 | 0.599 | 0.642 | 0.605 | 0.632 | 0.484 | 0.333 | 0.350 |
| Image creation & medium | **0.886** | -* | **0.805** | -* | **0.795** | -* | **0.589** | -* |
| Low-level characteristics | 0.674 | 0.624 | 0.635 | 0.660 | 0.566 | 0.457 | 0.402 | **0.369** |
| Problem strategy & solution | 0.758 | **0.719** | 0.660 | **0.740** | 0.716 | 0.539 | 0.406 | 0.307 |
| Writing & labels | 0.711 | 0.606 | 0.647 | 0.620 | 0.615 | 0.499 | 0.338 | 0.216 |

Table 8: Comparison of model performance across various question types for GPT4o, Claude3.5 Sonnet, Gemini1.5 Pro, and Llama3.2-11B V. The evaluation includes the average scores from our LLM evaluator across QA pairs generated synthetically by GPT4o and Claude3.5 combined (🤖) or by teachers (👩‍🏫). Examples of each question type listed above can be found in Table 6. The **max** score is bolded and the min is underlined across each QA and VLM. *For teacher-written QA, this question type had too few examples for robust performance estimates.

QA, and 25% are GPT-4o-generated QA. We stratify sample examples across all four VLMs. Then, annotators complete two tasks. First, given a question and a VLM's answer, we ask: *Is the provided answer correct?* Second, given the gold answer and the VLM's answer, we ask: *Do these two answers match?* Full instructions can be found in Appendix D.2. We ask these questions to identify cases where VLMs give correct answers that differ from gold standards, which we find only occurs in 36 out of 500 examples (7.2%).

**Assessing Our Automatic Metrics.** We compute Spearman correlations between automatic and human estimates of models' performance across teacher-, Claude-, and GPT-4o-generated QA sets and models, and find that LLM-based judgements are most similar to that of humans ($\rho = 0.801$), followed by ROUGE-L ($\rho = 0.472$) and then BERTSCORE ($\rho = 0.348$). In addition, across all 500 human-annotated model responses, binarized LLM-based judgements achieve a high accuracy of 0.896 and F1 score of 0.907 with respect to matching the human judgment.[6]

**Results and Findings.** We observe a range of performance across VLMs, most notably a gap between Llama 3.2 compared to closed-source alteratives (Table 7). BERTSCORE and ROUGE-L are able to differentiate models when judging synthetic QA, but they are less able to do so with teacher-written QA. According to our LLM-based evaluator, all three QA sets rank models similarity. In addition, questions pertaining to the correctness and errors tend to be most challenging for models, across both synthetic and teacher-written QA (Table 8). Thus, though synthetic QA can be noisy (§4), it can illuminate some differentiation of models' abilities.

Our human evaluation of models' responses surfaced a few additional observations around why and how models made errors. One common error involved models not being able to interpret dark images, even though their contents were visible to human annotators. Interestingly, we also found cases where models would answer a question correctly mathematically, but incorrectly with respect to the students' response. For example, to the question *Which whole number corresponds to 18/6 on the number line?*, all VLMs responded with *3*, even though the students' number line shows 18/6 aligned with 2. Altogether, the wide range of questions and student responses in our dataset can surface failure modes such as these.

## 7 Conclusion

Our work introduces a new dataset and benchmark, 🖼DrawEduMath, built upon teachers' annotations of K-12 students' handwritten responses to math problems. Overall, we hope our work will inspire further research for improving VLMs' capabilities in interpreting and supporting students' math learning in diverse real-world educational settings.

## 8 Limitations

**QA Quality and Utility.** Our paper involves the lengthy and careful collection of data from teachers, with the goal of creating a benchmark to assess VLMs' abilities to interpret students' handwritten

---

[6]Generally, false positives ($n = 46$) are more common than false negatives ($n = 6$).

work. However, every benchmark has a ceiling, and ours is no exception. The synthetic QA we created from teachers' descriptions can contain errors (§4), and ensuring that teachers' annotations are completely typo-free would require additional rounds of time-intensive proofreading. In addition to these issues, we made two qualitative observations that speak towards potential limitations of ⊞DrawEduMath for assessing models' visual understanding of students' handwritten work. First, we observed that some questions extracted from teachers' descriptions did not target content specific to the students' response, and instead may test for general mathematical knowledge, e.g. *What is a right angle?* Second, models' performance on some questions, such as the strategy the student used to solve a problem, should be weighed more heavily than performance on other questions, such as the type of paper used. We mitigate this concern by proposing a taxonomy of question types, to allow for more nuance than simply reporting model performance on aggregate. However, we encourage future work to aim for finer-grained categories to yield richer and more useful insights into model performance.

## 9   Ethical Considerations

**Risks and Harms of AI in Education.**   In the context of educational applications, AI models and systems may be viewed as inherently beneficial or for "social good." However, given the high-stakes nature of K-12 pedagogy, the deployment of VLMs, and AI generally, in education should carefully consider potential risks for harm [22]. For example, some pedagogical paradigms may have disproportionate influence on data availability and the design of technologies, thus perpetuating practices that may not cater towards a variety of learners [27]. We acknowledge that the images in our dataset, which is based on U.S.-centric Common Core math problems, may not cover the many varied ways in which students practice or learn math. In addition, we advocate for co-design of evaluative resources with in-domain experts, such as the K-12 teachers in our work.

**Data Privacy and Use.**   Our research has been overseen by our Institutional Review Board (IRB). Since some students' images might have PII (i.e., the students name might have been written on the piece of paper), we conducted extensive rounds of personally identifiable information (PII) removal, detailed in §3.1. ASSISTments, an online teaching platform, has a history of publishing data (with PII removed) from the platform for academic use [18]. We coordinated closely with ASSISTments, the license owner of the images, to establish clear boundaries on data usage and to develop our public release strategy.

## 10   Acknowledgements

We would like to thank Doug Jaffe, Laurence Holt, and Cristina Heffernan for their valuable feedback on the project. We would also like to thank some of our funding from NSF (1931523) and, IES (R305N210049 and R305T240029), the Jaffe Foundation, the Bill & Melinda Gates Foundation, and the Tools Competition.

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

# A    Annotation Details

## A.1    First Round

Figure 4 shows our data collection interface. Our instructions state:

*Instructions: Please describe out loud the Student Response on the right side of each image.*

*The Problem is provided on the left for context. If the Student Response is for a subproblem of a problem, the subproblem will be contained in a red box.*

*If you encounter issues that severely affect the quality of your recording, write "rerecord" in the Notes space so we mark it for re-annotation.*

*Press "Record" to start your recording.*

In addition to the description of the image, we ask teachers to answer two binary yes-no questions: *Is the Student Response too blurry or unreadable?* and *Does the Student Response include sensitive or personally identifiable information? Examples of this information include students'/teachers' names, emails, parts of people's hands/faces, or parts of homes/classrooms.* Out of 2,376 annotated images, 334 images were deemed too blurry and 4 images were removed by the secondary PII check. Other descriptions were not included in our final set of 2,030 due to transcription errors and annotation mistakes marked by teachers themselves.

The interface shown in Figure 4 evolved over the course of our two-month annotation period. After one week of annotations, we added the blurriness and PII questions so that teachers could communicate such properties via the interface instead of messaging project authors. In addition, we added a timer at the bottom of the page to track how long each annotation took, and added a notes box underneath the image. Initially, teachers were asked to describe all images out loud and submit a recording. Three weeks after starting annotations, we gave teachers the option to either record or type their description in the provided text box. Teachers requested this flexibility because they sometimes annotated in noisy environments. All recordings were transcribed automatically using OpenAI's Whisper [40].

## A.2    Second Round

### A.2.1    Writing Problem-specific Questions

For writing problem-specific questions, we redesign our data collection website from Appendix A.1 with a different set of instructions:

*Instructions: The image below shows a math problem. If there are multiple problems in the image, the focus on the one boxed in red.*

*What are some questions a teacher may ask about students' responses to this problem?*

*Propose **five** or fewer questions. Write **one question per line**.*

*Questions should be **self-contained**. If you want to add follow-up questions to a question, try to write those follow-ups as standalone questions, if possible.*

*Questions you ask might target:*

- *Words and numbers in the image (e.g., what labels are on the student's number line?)*
- *Lines and shapes drawn (e.g., did the student redraw the triangles shown in the problem?)*
- *Mathematical concepts (e.g., what kind of model is drawn in the image?)*
- *The student's approach (e.g., did the student use the standard algorithm?)*
- *Common errors that may arise (e.g. did the student _____ correctly?)*

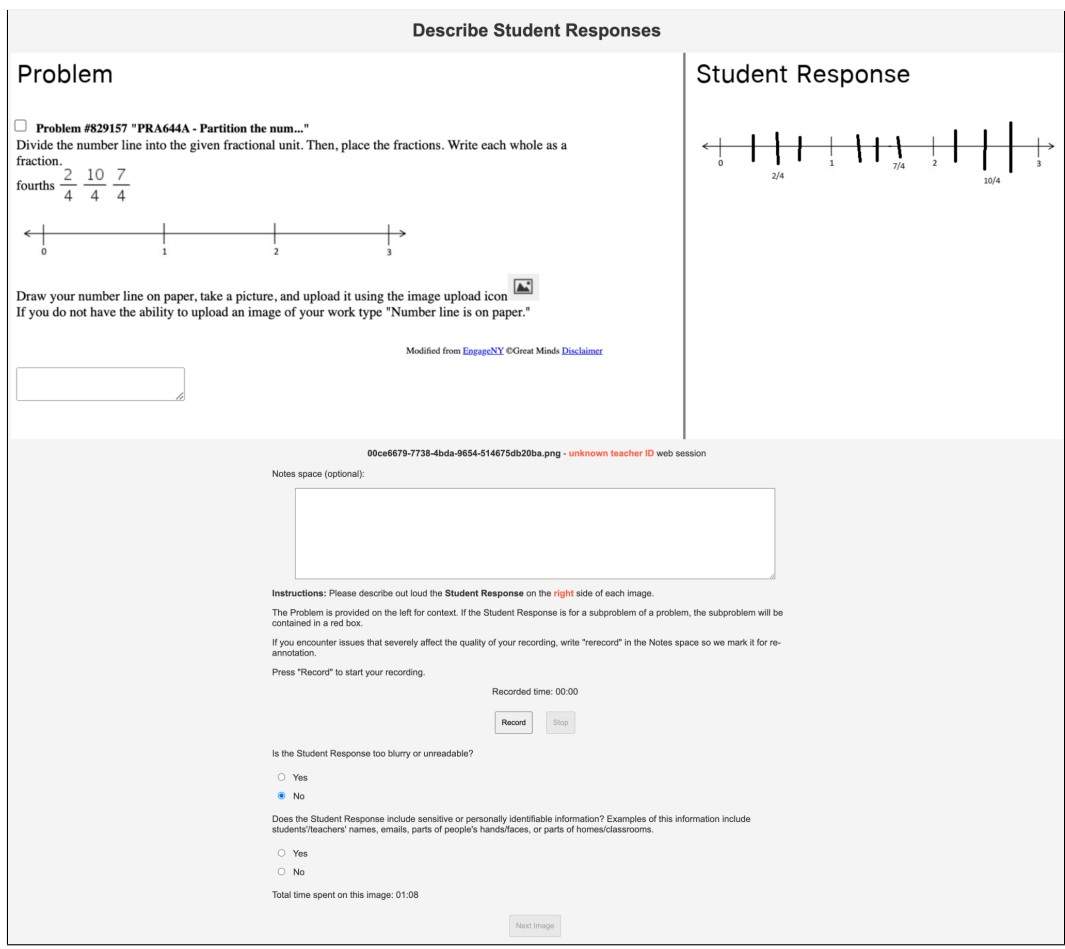

Figure 4: A screenshot of our recording website, where teachers would view an image from our dataset and either write or record a description of the student's response. Typically, "unknown teacher ID" would include the currently annotating teacher's ID.

Then, we present teachers a text box to in which they may write their questions. There is no audio recording option in this annotation step. Teachers can see the total time they have spent so far on a problem image at the bottom of the page, like they did in the first phase.

### A.2.2 Revising Annotations and Answering Teacher-written QA

Figure 5 shows what our annotation interface looks like for revising image descriptions and answering teacher-written QA. Our instructions state:

*Below is a description of the student's response written or spoken by a teacher. You may edit this description to correct any information that does not match the image.*

`{Text box}`

*Use the image of the student's response to answer the following questions in **full sentences**. Please **rephrase the question in your answer**, so that it is understandable without knowing the original question. **Scroll** to view more questions, as well as the option to add more questions & answers.*

At the end of the list of questions, four additional boxes were available for teachers to optionally add two image-specific questions and answers (one box for the question, one box for the answer, two pairs of QA total).

Figure 5: A screenshot of the interface teachers used to write answers to teacher-written questions about students' responses. Typically, "unknown teacher ID" would include the currently annotating teacher's ID.

## B Transforming Descriptions to QA Pairs

The first step in converting teachers' descriptions of students' responses into VQA pairs is decomposing the teacher-written captions into "facets", which are atomic descriptions of the information in the caption. Figure 6 shows our instruction prompt for the GPT4o and Claude 3.5 Sonnet, which converts teacher-written annotations into atomic facets or topics. The prompt follows a few-shot strategy, providing an example of a teacher-written caption and a list of atomic topics derived from it. The examples used in the prompts were curated with the help of an expert teacher.

For QA pair generation, the decomposed facets were again passed to the LLMs, prompting them to convert each facet into a QA pair. The prompt for this conversion is shown in Figure 7. Like the facet decomposition process, the prompt uses a few-shot strategy, providing examples of facets and their corresponding QA pairs, curated with the help of an expert teacher.

We map questions to question types using the prompt shown in Figure 8.

## C Model Benchmarking and Evaluation Details

Four vision language models (VLMs), GPT-4o, Claude 3.5 Sonnet, Gemini 1.5 Pro and Llama 3.2-11B Vision Instruct, were evaluated on their ability to interpret images of students' handwritten responses using developed QA pairs. Each model was prompted with an image of a student's response

**Decomposing captions to atomic facets**

```
You are given a caption describing a student's handwritten math image.  This
caption is a paragraph long description about the image.  Decompose this caption
into a list of atomic descriptions/facets, where each atomic description/facet
is about only one salient aspect of the image.  Each atomic description/facet
should be self-contained and capture only one idea from the caption.  One atomic
description/facet should not be a part of another atomic description/facet.
Anyone reading the atomic description/facet should be able to understand the idea
without needing to read the entire caption.  The atomic descriptions/facets are
short sentences or clauses extracted, but not inferred, from the given caption.

Output your answer as a list of strings.

For example, given this caption :
{ Example of a teacher written caption}
Generate :
{ Example of a list of atomic facets}

Decompose this caption:  {caption}
```

Figure 6: Prompt for decomposing teacher-written captions for images into atomic facets.

to a math problem and asked to answer a question from the generated QA pairs. The prompt used for generating answers based on the handwritten responses is shown in Figure 9.

For the evaluation of these models, five authors evaluated a random sample of 500 questions paired with the students' handwritten images, comparing the model's answer with the teacher's. The evaluation focused on: (i) the accuracy of the model-generated answer to the handwritten student response, and (ii) the similarity between the teacher-provided and model-generated answers.

To scale up the evaluation, we employed an LLM to assess the similarity of answers. We prompted the Mixtral 8x22B model to compare the two answers and provide a similarity score on a Likert scale. The prompt used for this evaluation is shown in Figure 10. Additionally, two automated metrics, BERTScore and ROUGEL were used to compare the answers.

## D Human evaluation

### D.1 Synthetic QA Quality Assessment

When assessing the quality of QA pairs are as follows, annotators are asked to select one bullet for each task below. The `numbers` in parentheses accompanying each answer choice indicate the total number of times that option was chosen by annotators across 100 QA pairs. This assessment step was done by asking annotators to download Markdown files containing one image and QA pair each, and mark `x` in checkboxes.

*Task 1: Can this question be answered by the provided image?*

*Q:* `a sampled question` $\mathcal{Q}$

*Response 1:*

- *Yes, the information in the images is sufficient to answer the question* (85)
- *No, the information in the images is not necessary to answer the question* (6)
- *No, the question is not answerable* (9)

*Task 2: Is the provided answer correct?*

*Q:* $\mathcal{Q}$

*A:* `the answer to` $\mathcal{Q}$

*Response 2:*

- *Yes, if an AI model returned this, I would trust it.* (82)

**Conversion of atomic facets/topic to QA pairs**

```
You are given a caption describing a student's handwritten math image.  You
are also given a list of short atomic descriptions derived from this caption.
Your task is to generate as many question-answer pairs as you can, with each
question focusing on a different atomic description from the provided list.  Each
question must be directly relevant to its corresponding atomic description and
self-contained, meaning that it should be answerable using only the information
provided in that specific atomic description.  Ensure each question is clear,
concise, specific and unambiguous.  Provide answers that are concise, and
directly address the content of each atomic description.  Avoid open-ended or
vague questions, and questions that can have multiple correct answers.
To avoid having questions with multiple correct answers possible, frame a
question as an alternative question with two mutually exclusive options, one
of the options being the answer.
Eg:  Instead of generating open ended question as this: ''Where is the purple
dot?'', generate close ended question such as: ''Is the purple dot to the left or
right of the number line'' , and instead of generating:  ''What type of content is
in the image?'', generate:  ''Is the content in the image hand-drawn or digital?''

Output your result as a list of JSON objects in the following format:
[{"question":  ..., "answer":...}, {"question":  ..., "answer":...},...]

For example,
Given the caption:
{ Example Caption }

And the atomic descriptions:
{ Example of a list of atomic facets }

Generate:
{ Example of a list of QA pairs }

Generate question and answer pairs given this image caption:{caption} and the
list of atomic descriptions:  {facets}
```

Figure 7: Prompt for converting atomic facets to QA pairs.

- *Maybe, but could be better. If an AI model returned this, I'd tolerate it but still have doubts.* (8)
- *No, I can see it trying but it's wrong. If an AI model returned this, I would distrust it.* (9)
- *No, this is just irrelevant/weird.* (1)

In the main paper, we binarize the responses to Task 1 by treating the first two options above as "Yes" and the third as "No" to separate out answerable and unanswerable questions. We also binarize Task 2's responses, by grouping "Yes" with "Maybe" and the two "No" together.

### D.2   Evaluating Model Performance

We verify the utility of our automatic evaluation metrics as well as their ranking of models by evaluating 500 model responses. Five annotators responded to the following questions in Markdown files containing images. Note that Response 1 below has options similar to Response 2 in Appendix D.1.

*Task 1: Is the provided answer correct?*

*Q:* a sampled question $\mathcal{Q}$

*A:* a model $\mathcal{M}$'s answer to $\mathcal{Q}$

*Response 1:*

- *Yes, if an AI model returned this, I would trust it.*
- *Maybe, but could be better. If an AI model returned this, I'd tolerate it but still have doubts.*
- *No, I can see it trying but it's wrong. If an AI model returned this, I would distrust it.*
- *No, this is just irrelevant/weird.*

*Task 2: Do these two answers match?*

*Q:* $\mathcal{Q}$

*A (Teacher):* `Gold answer to` $\mathcal{Q}$

*A (Model):* $\mathcal{M}$`'s answer to` $\mathcal{Q}$

*Response 2:*

- *Basically the same answer*
- *Similar but not same answer*
- *Neither similar nor different, not sure*
- *Quite different answers*

We binarize the above responses in Task 1 into "correct" and "incorrect" by grouping "Yes" with "Maybe" and the two "No" together. Similarly, we binarize the responses to Task 2 by grouping "Basically the same" and "Similar" together, and grouping "Neither" and "Quite different" together.

**Categorizing questions into question types**

You are categorizing questions related to assessing and understanding images of students' responses to math problems. You will receive a list of question types lettered A to H, including examples of questions that fall within each type. Your task is to assign an unlabeled question to a letter representing a question type.

Here are all possible question types:
A. Questions around how the image or its contents were created, such as medium or paper type. Examples: "Are the rectangles in the image hand-drawn or computer-generated?", "Is the image of handwritten student work on a whiteboard or on paper?", and "Is the student's handwriting on lined paper or blank paper?".
B. Questions focusing on writing or labels in the image. Examples: "What is the top of the rectangles labeled with?", "Are the x values from left to right 24, 48, 72, 96, and 108 or 24, 48, 72, 94, and 100?", "Are the disks on the board numbered or unnumbered?", "Are every consecutive whole number labeled on the y-axis or only some numbers?", "What fraction is written above the number 1?", "According to the student's note, is the table harder or easier to use?", and "What equation is typed on the page?".
C. Questions inquiring about the low-level composition of drawings/diagrams, including the positioning of content. These questions should only require minimal understanding of math concepts. Examples: "Along the number line, has the student drawn tick marks?", "Which digit in 26 has the student circled?", "Are the lines completely straight or not entirely straight?", "What color is the shaded piece in the bottom strip?", "Are the dots arranged randomly or in groups?", "Are the vertical lines inside the rectangles equally spaced?", "Does the second arrow go from -6 to +6 or from +6 to -6?", and "In the place value chart, where does the student write the digit 7?".
D. Questions that involve enumerating visual content. Examples: "How many green dots are drawn in a row?", "What is the total number of cells in the table?", "According to the student's actual drawing, how many groups and how many dots are in each group?", and "Does the tape diagram drawn by the student have multiple sections or just one section?".
E. Questions that involve higher-level understanding of math shown in the student's response, including knowing what specific content is intended to represent. Examples: "What is the highest number on the tick marks?", "Are coordinates given in the image?", "Are the numbers below the line whole numbers or fractions?", "Which piece is shaded to represent 1 over 4?", "Are all the angles in the image acute or obtuse?", "3 garlic cloves correspond to how many tablespoons of olive oil?", "According to row 4, how much is charged for 6 lawns?", and "Is the purpose of this number line to show where to round 26 or where to round 25?".
F. Questions pertaining to the student's problem solving steps, strategy, or solution. Examples: "How does the student demonstrate the multiplication in the equation?", "What is the result of the butterfly method?", "To what number is the student estimating 2,803?", "What is the result of 8 divided by 2?", "According to the answer sentence, how many homework papers does Ms. McCarthy have left?", and "According to the diagram, how much do three-sevenths equal?"
G. Questions that judge the correctness of the student's work. Examples: "Does the student correctly or incorrectly identify the base of the prism?", "Does the student have any misconceptions regarding coordinate pairs?", and "Does the student put the decimal in the correct place in the product?".
H. Other

Your response must begin with a capital letter ranging from A to H. For example:
Question: Did the student correctly draw two rows in their array?
Category: G.

Now, assign the following question to a question type that it fits best. Remember to begin your response with a capital letter designating a question type.
Question: question
Category:

Figure 8: Prompt for categorizing questions into question types.

**Generating answer for a question about student's handwritten response**

```
You will be provided an image containing two parts:  a math problem on the left
side, and a student's handwritten response to that problem on the right.  Your
task is to answer a question about the student's work on the image's right side.
Your answer should be clear and concise.  If possible, provide short answers that
are five words or less.
Do not solve the problem yourself; just answer the question based on the
student's response in the provided image.  Focus on the student's work and not
on the problem that is provided on the left side.

For example,
Question:  "What equation is written above the diagram?"
Your answer:  "3x + 2 = 8"

Question:  "How many boxes are the width and length of the graph?"
Your answer:  "18 by 10"

Question:  "What is drawn on the grid?"
Your answer:  "A square"

Now, using an image of a math problem and student's response, answer the
following question.
{question}
{image}
```

Figure 9: Prompt used with VLMs for answering question about the student's handwritten response.

**Comparing model's answer with teacher provided answer**

```
Given, Question:  {question}
Answer 1:  {teacher_a}
Answer 2:  {model_a}

Rate the level of similarity between these two answers with respect to how well
they answer this question.  The Likert rating options are:
4.  Basically the same answer
3.  Similar but not same answer
2.  Neither similar nor different
1.  Quite different answers

Provide both the Likert rating followed with an explanation as to why they are
similar.  Format the output as a valid parsable JSON like:
{"rating":  3, "reason":  "Because..."}
```

Figure 10: Prompt used for comparing model-generated answer with teacher-provided answer about student handwritten responses.

