# OpenReview forum: "DrawEduMath: Evaluating Vision Language Models with Expert-Annotated Students’ Hand-Drawn Math Images"
_NeurIPS.cc/2024/Workshop/MATH-AI — MATH-AI 24_

### Official Review · Reviewer_xBbu · 2024-10-06
**A new dataset for proposed for mathematical problems**

**Rating:** 4
**Confidence:** 5

**Review:**

This paper proposes new dataset named DrawEduMath from the handwritten question and answers from the 188 K-12 math problems where the authors hired research assistant to this task. They have collected these questions answered by students from diverse question types in the mathematics. To get each question and answer from each image handwritten images collected the research assistants cropped each question and made the student identity anonymous. In the dataset they also have collected metadata of grade level, descriptions and topic categories.
Later they have evaluated the dataset on latest large language models GPT-4o, and Gemini -1.5 Pro and tabulated the results with best metrics that are used to evaluate the Large Language Models. In the evaluation the authors have found that they GPT-4o is performing better than the Gemini-1.5 Pro in all the metrics. In the end the authors have mentioned that they are going to improve the performance of annotations and they claim that this kind of dataset improves the capabilities of Vision Language Models in solving diverse mathematical problems.

---

### Official Review · Reviewer_1tfY · 2024-10-07
**Review of DrawEduMath: An Expert-Annotated Dataset of Students’ Math Images**

**Rating:** 6
**Confidence:** 5

**Review:**

This work introduces a new dataset of students' solutions to math problems along with synthetically generated QA pairs that extract details from the students' responses. The work is interesting as we need more benchmarks to understand how models perform in terms of understanding tutoring and performing tutoring in turn.

Although the work is promising in order to further our understanding of VLMs capabilities in tutoring, I find these areas for improvement of this work:

1) Authors can mention the implications of their work more clearly and also can motivate their work better. I find it challenging for me to understand the specific shortcomings in the models and how to use these works to make stronger models for tutoring purposes. I find interpreting Table 4. a bit challenging as the measures are comparing VLMs' responses to questions with another LLMs' summarized text from an expert tutor. We don't know what values are satisfactory for the real-world applications of VLMs. Although I agree with the general message of the papers that there are areas for improvement in current models, authors can try to distinguish different aspects of such shortcomings. For instance, do they struggle with arithmetic, understanding drawings, or something else? This distinction can be helpful in informing researchers in the field to focus on the specific shortcomings to overcome.

2) I am wondering if fine-tuning models such as Llava can improve their performance on this task. Adding fine-tuning to the evaluation strategies can significantly improve the significance of the paper.

3) As the authors have mentioned in the footnote on page 4, correlations between human judgments and automatic metrics are not high enough. (0.55 for LLMs, 0.12 and 0.15 for BERTScore and ROUGE respectively.) Given this, I am wondering how reliable the differences in Table 4 are. As an idea, the authors might want to decompose VLM responses into two sets of questions, factual ones (e.g., counting in Table 3.) and others. Many of the questions in their benchmark are factual questions, so perhaps a binary measure that checks for factual correctness in VLMs' responses might work better and be a more reliable metric.

4) It would be great if the authors could increase the number of QA pairs and problems in their dataset so it can be used for fine-tuning.


Overall, I find this a very interesting work, and support more benchmarks that have educational values and take pedagogy into account such as this work. I guess this work is promising to make contributions to the study of VLMs in educational settings.

---

### Official Review · Reviewer_aaBu · 2024-10-09
**This paper proposes a good benchmark for VLMs, which may have good potentials for related future works.**

**Rating:** 7
**Confidence:** 4

**Review:**

Thank you for submitting this notable paper. This work proposes a new benchmark (i.e., DrawEduMath) for the performance evaluation of vision language models (VLMs).

It is encouraging to see that the authors have been thoughtful and include multiple different types of information in each image datapoint, which makes sense.  A good comparison between Gemini-1.5 Pro and GPT-4o is included in the paper.

Overall, I could see good potential to extend the scale and refine the datapoints to make the benchmark even stronger. So, I would recommend to accept this paper.

---

### Decision · Program_Chairs · 2024-10-09

Accept